# Tocilizumab for the Treatment of Familial Mediterranean Fever—A Randomized, Double-Blind, Placebo-Controlled Phase II Study

**DOI:** 10.3390/jcm11185360

**Published:** 2022-09-13

**Authors:** Joerg C. Henes, Sebastian Saur, David M. Kofler, Claudia Kedor, Christoph Meisner, Marion Schuett, Martin Krusche, Ina Koetter, Theodoros Xenitidis, Hendrik Schulze-Koops, Eugen Feist

**Affiliations:** 1Centre for Interdisciplinary Clinical Immunology, Rheumatology and Auto-Inflammatory Diseases and Department of Internal Medicine II (Oncology, Hematology, Immunology and Rheumatology), University Hospital Tuebingen, 72076 Tuebingen, Germany; 2Division of Rheumatology and Clinical Immunology, Department I of Internal Medicine, University Hospital Cologne, 50441 Cologne, Germany; 3Institute of Medical Immunology, Charité Universitaetsmedizin Berlin, 10115 Berlin, Germany; 4Study Center, Robert Bosch Hospital, 70376 Stuttgart, Germany; 5Institute for Clinical Epidemiology and Applied Biometry, University Hospital Tuebingen, University Tuebingen, 72076 Tuebingen, Germany; 6University Hospital Hamburg Eppendorf, 20095 Hamburg, Germany; 7Division of Rheumatology and Clinical Immunology, Department of Internal Medicine IV, Ludwig-Maximilians-University of Munich, 80539 Munich, Germany; 8Helios Department of Rheumatology, Cooperation Partner of the Otto-von-Guericke University, 39245 Gommern, Germany

**Keywords:** Familial Mediterranean Fever, tocilizumab, interleukin 6, colchicine resistant, treatment

## Abstract

Background: The purpose of this trial was to evaluate the effectiveness and safety of the IL-6 receptor antibody Tocilizumab (TCZ) in the treatment of Familial Mediterranean Fever (FMF). Methods: This was a randomized, double-blinded, placebo-controlled phase II trial in adult patients with active FMF and an inadequate response or intolerance to colchicine (crFMF). The physician’s global assessment of disease activity (PGA), based on a five-point scale for six symptoms, was used as a clinical score, which had to be >2 at screening, together with elevated c-reactive protein (CRP) or erythrocyte sedimentation rate (ESR) and serum amyloid A (SAA) levels, to be eligible for inclusion. Patients were randomized 1:1 to either receive monthly TCZ or a placebo over a period of 24 weeks. The primary endpoint was the number of patients achieving an adequate response to treatment at week 16, defined as a PGA of ≤2 and normalized ESR or CRP and normalized SAA. Results: We randomized 25 patients with a median age of 31 years. At week 16, an adequate treatment response was achieved by two patients in the TCZ and none of the patients in the placebo arm (*p* = 0.089). SAA levels normalized with TCZ, but not with the placebo (*p* = 0.015). Conclusion: In this first randomized, placebo-controlled study in patients with active crFMF, more patients in the TCZ arm experienced a response to treatment in comparison to those receiving the placebo. As the prevention of amyloidosis is a major treatment goal in FMF, the normalization of SAA in TCZ-treated patients is essential. These findings have to be confirmed in a larger trial.

## 1. Introduction

Familial Mediterranean Fever (FMF) is the most common monogenetic auto-inflammatory disease (AID). It is characterized by chronic inflammation without autoantibodies or antigen-specific T-lymphocytes. Patients typically suffer from recurrent episodes of fever, abdominal and thoracic pain due to serositis, skin irritations, and arthralgia. In addition to their reduced quality of life [1], FMF patients are at risk for severe amyloidosis as a long-term complication, which can lead to organ failure and increased mortality. Amyloidosis is characterized by the abnormal deposition of insoluble proteins in various tissues and organs and is a known severe complication of uncontrolled inflammation, e.g., due to AID. The worldwide prevalence of FMF is estimated at 100,000–150,000 individuals, with increasing reports from countries other than the typical Mediterranean countries [2,3,4].

The treatment goals in FMF are to prevent patients from clinical attacks and to suppress chronic subclinical inflammation and, thus, the risk for amyloid A (AA) amyloidosis [5,6,7].

For a very long time, colchicine was the only approved medication to reduce FMF attacks, and until today, it was the only medication that showed a reduction in the rate of amyloidosis in FMF patients. In 10–15% of patients, colchicine is not adequately effective or the patients suffer from intolerance [5,6,7]. These cases are called colchicine-resistant FMF (crFMF) if the patients suffer from ongoing attacks (usually ≥ 1 per month), despite the maximum tolerated dose of colchicine [7].

The cornerstone of the pathophysiology in FMF is the inappropriate production of proinflammatory cytokines, such as interleukins (IL)-1 and 6, due to mutations of the Mediterranean Fever (*MEFV*) gene located on chromosome 16 [8].

Antagonists of IL-1 (anakinra, canakinumab, and rilonacept) have been shown to be effective in these patients and are approved for FMF in Europe and the USA (rilonacept only in USA) [9,10,11,12,13]. Up to now, these antagonists could demonstrate effective treatment of FMF attacks and improvement of kidney function in patients with amyloidosis, but there is no evidence yet for reducing the incidence of amyloidosis. Therefore, other treatment strategies are still needed.

Tocilizumab (TCZ) is a monoclonal antibody directed against the IL-6 receptor and is approved for treating rheumatoid arthritis, systemic juvenile idiopathic (poly)arthritis, giant cell arteritis, and cytokine release syndrome, and it has a favorable safety profile. Several case reports and series have revealed promising results from treating crFMF attacks and amyloidosis with TCZ [14,15]. Therefore, we aimed to investigate the efficacy and safety of TCZ in crFMF patients in a randomized controlled trial.

## 2. Materials and Methods

### 2.1. Study Design and Patient Selection

The TOFFIFE study was a randomized, double-blinded, placebo-controlled trial conducted at five German centers (for the list of centers, please see Appendix A) specializing in autoinflammatory diseases.

The target population for this study included adult (≥18 years) patients with FMF, according to the Tel-Hashomer criteria (see Appendix A) [16], who had at least one (heterozygous or homozygous) known pathogenic or likely pathogenic mutation of the *MEFV* gene and active disease with an inadequate response or intolerance to colchicine. Active disease was defined as having had at least one attack of fever (>38.0 °C) and/or pericarditis and/or serositis and/or testicular involvement and/or arthritis and/or erysipelas-like rash, despite the maximum tolerated dose of colchicine within 12 weeks prior to the baseline visit. In addition, elevated levels of C-reactive protein (CRP > 0.5 mg/dL) and/or erythrocyte sedimentation rate (ESR > 20 mm/h) and/or serum amyloid A (SAA > 10 mg/dL) had to be present at screening. The physician’s global assessment of disease activity (PGA), based on a five-point scale (from 0 to 4, with 0 = no symptoms during the last 4 weeks, 1 = minimal (defined as at least one of the above mentioned symptoms < 1 week out of the last 4 weeks), 2 = mild (symptoms < 2 weeks out of the last 4 weeks), 3 = moderate (symptoms < 3 weeks out of the last 4 weeks), 4 = severe disease (symptoms every day during the last 4 weeks), according to the definition used in the CLUSTER study [11]) and associated with six clinical signs and symptoms (which orientated on the auto-inflammatory diseases activity index (AIDAI) [17]; total range: 0–24) was used as a clinical score to evaluate disease activity throughout the study (see Appendix A for further PGA definition). The PGA score had to be >2 at screening to randomize patients.

Patients had to be excluded if they had ever been treated with TCZ, were on glucocorticosteroids (GC) > 10 mg/day at baseline, or were on any biologic or non-biologic immunosuppressive treatments. Different wash-out periods had to be respected. Patients with severe accompanying diseases or other autoimmune/autoinflammatory diseases were excluded (for a complete list of inclusion/exclusion criteria, see Appendix A).

Colchicine was continued at the latest tolerated dose throughout the study. Non-steroidal anti-inflammatory drugs (NSAIDs) and antipyretic treatment were allowed as a rescue therapy for attacks.

### 2.2. Randomization, Blinding, and Procedure

Patients were allocated 1:1 by a computer-generated block randomization using randomly chosen variable block sizes (4, 6, or 8) to either receive monthly TCZ intravenously with 8 mg/kg bodyweight (a maximum of 800 mg) or a placebo infusion blinded to patients and treating physicians by cover. The medication/placebo was prepared by an unblinded member of the study group at each center or by the local pharmacy. As both PBO and TCZ are transparent solutions, a simple cover was chosen to blind the patients and physicians. The subjects were followed in an outpatient setting. After their screening visit (maximum 28 days prior to baseline) and randomization, the subjects were treated and evaluated at the study center at day 0/baseline and at weeks 4, 8, 12, 16, 20, 24, 28, and 32/end of evaluation. Patients with an inadequate response, in the opinion of the treating physician, had the opportunity to receive open label TCZ for the weeks 16, 20, and 24. For the exact study scheme, see Figure 1.

### 2.3. Endpoints

The primary endpoint was the number of patients achieving complete remission at week 16, defined as a PGA score of ≤2, a normalized ESR or CRP (the item that led to inclusion had to normalize), and normalized SAA.

The secondary endpoints were the efficacy of TCZ as a rescue therapy, measured by PGA score at week 28, the proportion of patients with a serological remission (defined as a CRP of <0.5 mg/dL or normalized SAA level) at weeks 16 and 28, the improvement of patients, and the physician’s global assessment of FMF activity on a visual analogue scale of 0–10, as well as the possibility of reducing the intake of NSAIDs or the impact of disease on health-related quality of life, as assessed by the German “Funktionsfragebogen Hannover” (FFbH) scale and the overall safety assessed by adverse events of TCZ in patients with crFMF. Patients were asked to fill out a patient’s diary (see Appendix A) to record signs and symptoms, as well as NSAIDs intake, to help the investigators judge the PGA.

### 2.4. Statistical Analysis

This study was a phase-II-trial designed to demonstrate the superiority of TCZ compared to a placebo. The sample size determination was based on the assumption that the predicted proportion of patients experiencing a primary endpoint in the placebo group was 40%. With 15 patients in each group, a Fisher’s exact test has a power of 80% to show an increase of the endpoint proportion in the TCZ group of up to 83% or higher. For this phase-II-study, a two-sided significance level of *p* < 0.2 was chosen. The statistical analysis was pre-defined in a statistical analysis plan, which was written and approved before unblinding. The proportion of patients who met the primary endpoint were compared and statistically assessed using a two-sided Mantel–Haenszel chi-square test to test the null hypothesis of equal proportions in the two therapy groups. For the primary endpoint, patients were analyzed according to the treatment that they were assigned to at randomization (ITT). Missing endpoints due to missing data were imputed as non-responders for the confirmatory analysis in order not to overestimate the results. Secondary endpoints were compared and statistically assessed using two-sided Mantel–Haenszel chi-square tests, stratified by center (PGA, SAA, and CRP), analysis of covariance with baseline values as covariates (VAS and FFbH), or Fisher’s exact test (NSAID intake).

This study is registered with ClinicalTrials.gov (NCT03446209) and was approved by the ethics committee of the University of Tuebingen and all participating centers. The written informed consent of all patients was obtained.

## 3. Results

Of the 31 included patients, 25 were randomized. Six patients dropped out before randomization due to screening failures, missing data, withdrawal, or incompliance (for the consort flow chart, see Appendix A). For further evaluation, the intention to treat (ITT) population consisted of 25 crFMF patients with a median age of 31 years (range 18–53 years), of which 14 (56%) were female. The median disease duration was 16 years (range 0–44) and 56% (*n* = 14) had a homozygous and 44% (*n* = 11) a heterozygous (of which 64% (*n* = 7) had two heterozygous mutations) mutation of the *MEFV* gene (for details on mutations see Appendix A). In their medical histories, all patients were treated with colchicine, with an inadequate response, whereas in three patients, anakinra, and in one patient, canakinumab was ineffective or not tolerated before entering the study. Biologicals were stopped at least 12 weeks before entering the study. GCs and NSAIDs were used in 32% (*n* = 8) and 64% (*n* = 16), respectively. GC (median dose of 5 (0–10) mg/day) and colchicine doses (median dose of 1.6 (0–3.5) g/day) were stable throughout the study. For baseline characteristics of the study population, see Table 1.

At week 16, the timepoint for the primary endpoint, two (15.4%) patients in the TCZ arm reached the primary endpoint with a PGA score of ≤2 and normalization of SAA and CRP and/or ESR, while none of the patients in the placebo arm did so. With respect to the small number of patients and the pre-specified significance level of α = 0.2, the superiority of TCZ compared to the placebo was arithmetically given (*p* = 0.089, Appendix A), but it has to be interpreted with cautiousness.

Due to insufficient response or side effects, seven (28%) patients terminated earlier than week 16 and were counted as drop-outs, which led to missing data at week 16. These early terminations occurred in four (33.3%) patients in the PBO arm and in three (23.1%) patients in the TCZ arm. In the TCZ arm, this was due to ineffectiveness in two patients and side effects in one patient. They were included in the intention to treat analysis.

For the detailed results of the primary and secondary endpoints and their limitations, see Table 2.

The difference in PGA was not significant (see Figure 2). SAA and CRP levels normalized with TCZ but not with PBO (Table 2 and Figure 3; for further details, see Appendix A). This difference between TCZ und PBO was very clear; for SAA, *p* < 0.015 and for CRP, *p* < 0.011.

No difference was seen in the reduction of NSAIDs intake (PBO yes = 3, no = 4; TCZ yes = 2, no = 5; missing: PBO = 5, TCZ = 6), nor was there a reduction of the patients’/physician’s visual analogue scale (VAS) or of the FFbH score, although, in both groups, a reduction was visible (Appendix A).

At week 28, with 17 remaining patients and after having had the opportunity for a rescue treatment at week 16, the responder rates (PGA score of ≤2 and the normalization of SAA, ESR, and/or CRP) were 25% (*n* = 1) in those patients who changed from the placebo to TCZ (*n* = 4) and 20% (*n* = 2) in those patients who continued with TCZ (*n* = 10). Of note, all three patients remaining on PBO were non-responders (*p* = 0.642). CRP and SAA normalized in 75% (*n* = 3) and 50% (*n* = 2) of the patients, respectively, after changing to TCZ.

### Safety

Until week 16, 73 adverse events occurred in 20 patients (Table 3), of which five (6.8%) in three patients were serious (one SAE (2%) in the TCZ arm and four SAEs (18.2%) in two patients of the PBO arm). The one SAE in the TCZ arm was ileitis. Severe disease activity in three cases and a car crash with hospitalization were seen in the PBO arm. All SAEs resolved after specific treatment. One AESI (increased liver enzymes) occurred in the TCZ arm. Most AEs were mild and 19% were associated with FMF flares. Given the relatively small number of patients, no other safety aspect could be evaluated, but no serious infection, life-threatening events, opportunistic infection, deaths, or pregnancies occurred during the entire study.

After week 16 and until the end of study visit, 35 more AEs were reported (Appendix A), of which four were SAEs: one in the ongoing PBO group (psychiatric disorder), two in the TCZ arm (two in the same patient with an FMF flare), and one (rectal bleeding due to hemorrhoids) after changing from PBO to TCZ.

## 4. Discussion

FMF is a disease with a high burden due to recurrent episodes of fever and pain, reduced quality of life, and the risk of developing end-stage organ failure due to amyloidosis. This is the first randomized, placebo-controlled study on the efficacy and safety of the IL-6 inhibitor tocilizumab in patients with active disease and despite colchicine treatment.

The phase II trial met the primary endpoint to demonstrate the advantage of TCZ over PBO, although only a small numerical difference was found. It is noteworthy that the proportion of patients with a good response to TCZ was lower and the drop-out rate was higher than expected, which clearly lowers the significance and the evidence of these findings.

Our response criteria were very strict, as patients had to achieve a complete remission with a PGA score of ≤2 and a normalization of the inflammatory parameters (CRP/ESR and SAA). This required no, or only the mildest, symptoms during the 4 weeks preceding the primary endpoint. No PBO-treated patient reached the primary response criteria for successful treatment. Nevertheless, most of the PBO-treated patients also achieved a reduction of their PGA score during treatment, leading to the perception of a strong influence of close and intense care on the wellbeing of our FMF patients. This could be related to the close medical care during this phase and the associated subjective security and stress reduction. This low activity level under PBO treatment was not expected when writing the protocol, owing the lack of bigger randomized, placebo-controlled trials in FMF thus far. The biggest trials in crFMF with IL-1 antagonists have included only very few patients. For anakinra, canakinumab, and rilonacept, only one randomized controlled trial with 25, 63, and 14 patients [11,12,13], respectively, have been published. These studies were published after composing this study protocol. Rilonacept is still not available in Europe. The two studies with anakinra and canakinumab experienced the same recruitment problems that we did.

The definition of an ideal primary endpoint is still missing. We decided for the PGA score as it seemed more flexible than simply counting the number of attacks, such as in the anakinra RCT [12] and orienting it at the AIDAI score for the diary and definitions from the CLUSTER trial with canakinumab [11]. At least in the CLUSTER trial, which was a basket trial with three different autoinflammatory diseases included, the PGA score could discriminate quite well between canakinumab and the placebo. Nevertheless, the definitions of PGA score were chosen differently and are thus not comparable. We determined for six typical FMF symptoms and graded from not present to present throughout the last 4 weeks; therefore, a sum score of up to 24 was possible. The CLUSTER trial used a PGA score with a total sum of 0–4. The response to treatment was defined identical in both studies, and patients had to achieve a PGA score of ≤2 to be counted as a responder. Here, the CLUSTER trial demonstrated the significant superiority of canakinumab over the placebo. The randomized controlled trial with anakinra compared the number of FMF attacks, and more patients treated with anakinra achieved a significant reduction of flares [12]. In both trials, not all patients responded well to the IL-1-blockade, still leaving the necessity of alternative treatment options.

Of major importance, in all three studies no statistically significant difference in the reduction of SAA or CRP could be found, although disease activity improved. The normalization of CRP was expectable in our study, as blocking IL-6 leads to downregulation of CRP production in the liver, and almost every elevated CRP level normalizes during treatment with TCZ. On the other hand, SAA levels only normalized in TCZ-treated patients, and this normalization is very important to our patients with regard to the prevention of organ amyloidosis. As one of our main treatment goals was the prevention of amyloidosis, these results raise the hope for further effectiveness. In the CLUSTER trial, several patients did not reach a normalization in SAA during therapy [11], and with anakinra and rilonacept, no considerable difference of final SAA levels between the anakinra and placebo groups was noted. Thus, at least with this secondary endpoint, TCZ seems to be even more effective than the IL-1 blockade, although IL-1 antagonists definitely remain as the first-line treatment option for crFMF with regard to its effectiveness and approval status.

No opportunistic infections and no deaths occurred, and infection rates were even lower in the TCZ arm, whereas the absolute number of adverse events was higher in the TCZ group. Most of the AEs were FMF flares, as were most of the SAEs in the PBO arm. One AESI occurred in the TCZ arm (an increase in liver enzymes, a known side effect of TCZ). Overall, the treatment was safe and no new safety aspects occurred.

Our study has limitations; in particular, the limited number of patients and the high drop-out rate reduce the significance of our findings. Recruitment of patients was very difficult and some centers could not include even a single patient. Overall, compliance to the protocol was poor. Some patients recalled their consent even before randomization, and others had already insisted on rescue therapy at week 8 or 12, which, overall, led to the reduced number of patients. Originally, we planned to have 30 patients, but only 25 were accessible for the primary endpoint and only 17 could be included in the analysis for week 28. For future trials, higher patient compliance must be achieved to improve the data quality. In addition, we would recommend changing to a cross-over study design as this may improve compliance. There are more precise laboratory markers for disease activity available, for example, S100, but at the time of the study, the protocol was written and this parameter was not routinely used in the participating centers. Colchicine resistance was recently newly defined, but this concrete definition was not available when the study was conducted.

## 5. Conclusions

In summary, this phase II randomized controlled trial showed first encouraging results for TCZ in crFMF, especially with regard to the reduction of SAA levels, although the sample size was small. A larger multicenter study that includes regions with higher FMF prevalence than in Germany, and perhaps including patients with a very high risk for amyloidosis, is therefore reasonable and necessary to further clarify the benefits of treatment with TCZ in patients with crFMF.

## Figures and Tables

**Figure 1 jcm-11-05360-f001:**
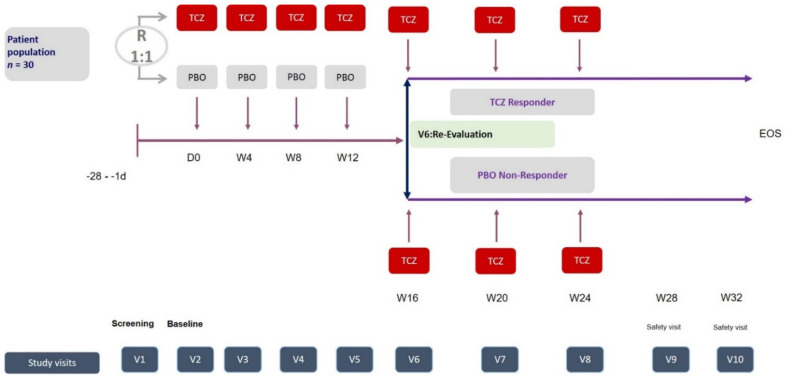
Study scheme with 1:1 randomization and rescue arm after week 12. Abbreviations: TCZ: tocilizumab, PBO: placebo; W: week, V: visit; crFMF: colchicine-resistant Familial Mediterranean Fever.

**Figure 2 jcm-11-05360-f002:**
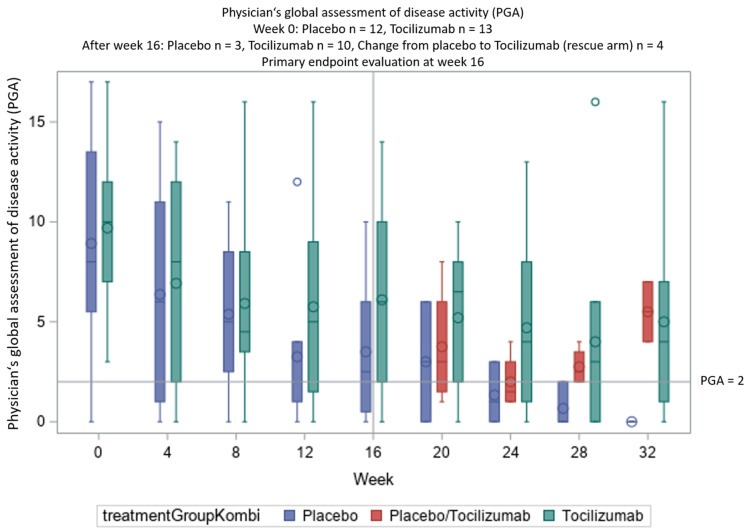
Secondary endpoint: the physician’s global assessment (PGA) over time, showing the response to treatment according to the PGA in both arms of the trial and the increase of disease activity after the treatment stopped.

**Figure 3 jcm-11-05360-f003:**
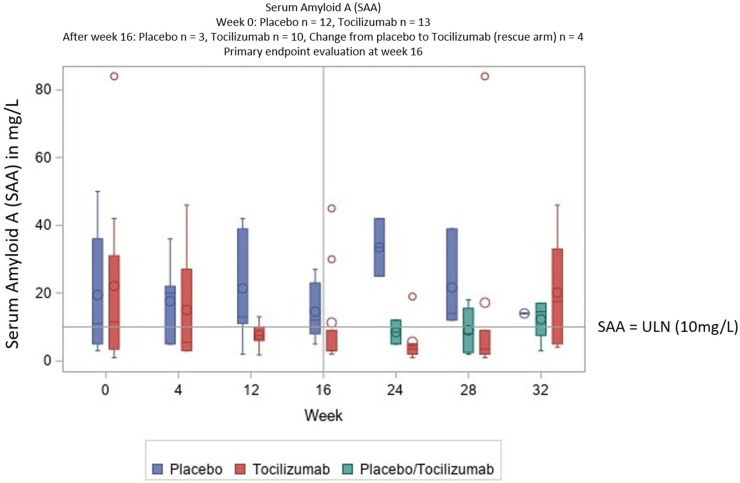
Secondary endpoint: the clear reduction of SAA in the TCZ but not in the PBO arm, and the rerise of SAA after discontinuation after week 28. A normal SAA value is ≤10 mg/L. The outliers of >100 mg/L were excluded in this graph (in total, in 12 patients, the SAA values over the period of 32 weeks were >100 mg/dL, with nine in the placebo group and three in the TCZ group).

**Table 1 jcm-11-05360-t001:** Patients’ baseline characteristics: there were no differences between the treatment groups in any characteristic. Abbreviations: FFbH= Funktionsfragebogen Hannover; DAS28 = disease activity score; VAS: visual analogue scale; PGA = physician’s global assessment; SAA = serum amyloid A; CRP = C-reactive protein; ESR = erythrocyte sedimentation rate; NSAID = non-steroidal anti-inflammatory drugs.

Baseline Characteristics	Placebo (*n* = 12) Median (Min–Max) *n* (%)	Tocilizumab (*n* = 13) Median (Min–Max) *n* (%)
Age, years	28.5 (18–41)	33 (18–53)
Female, *n* (%)	7 (58.3)	7 (53.8)
Disease duration, years	12.5 (0–29)	18.0 (2–44)
Homozygous mutation, *n* (%)	7 (58.3)	7 (53.8)
Baseline weight, kg	70.5 (61–117)	70 (50–140)
Baseline height, cm	170 (159–183)	166 (150–180)
Baseline FFbH, %	88.9 (8.3–100)	91.7 (66.7–100)
Baseline DAS28	3.2 (1.8–5.5)	3.4 (1.2–7.2)
Baseline VAS patient activity	47.0 (0–77)	58.0 (15–88)
Baseline VAS physician activity	66.5 (0–90)	60.0 (24–82)
Baseline PGA (0–24)	8 (0–17)	10 (3–17)
Patients with PGA ≤ 2, *n*	1	0
Baseline SAA (mg/L)	18 (3–275)	31 (1–661)
Patients with SAA ≤ 10 mg/L, *n*	3	4
Baseline CRP (mg/dL)	0.9 (0.2–4.2)	1.4 (0.0–15.3)
Patients with CRP ≤ 0.5 mg/dL, *n*	4	4
Baseline ESR (mm)	14 (0–41)	14 (4–65)
Patients with ESR ≤ 20 mm/h, *n*	10	7
Previous medication, *n* (%)		
Colchicine	12 (100)	13 (100)
Glucocorticosteroid	2 (16.7)	6 (46.2)
Nsaids	8 (66.7)	8 (61.5)
Anakinra	1 (8.3)	1 (7.7)
Canakinumab	0	1 (7.7)

**Table 2 jcm-11-05360-t002:** Primary and secondary endpoints: at week 16, more patients had achieved the responder criteria in the TCZ arm (ULN = upper limit of norm) (two-sided Mantel–Haenszel chi-square test, stratified by center), and the missing values are imputed as non-responders.

**Responder Variable**	**Placebo** **(*n* = 12) (%)**	**Tocilizumab** **(*n* = 13) (%)**	**Total** **(*n* = 25) (%)**	
**Responder at Week 16 (Missing Values Imputed as Non-Responders)**				
Responder (PGA ≤ 2, SAA, ESR, and/or CRP ≤ ULN)	0 (95% CI: 0–22%)	2 (15.4%) (95% CI: 2–45%)	2 (8.0%)	*p* = 0.089
Non-responder	12 (100%)	11 (84.6%)	23 (92.0%)	
The missing values (due to missing values, no responder evaluation at week 16) imputed as non-responders	4	3	7	
**Secondary endpoints**	**Placebo** **abs. (%)**	**Tocilizumab** **abs. (%)**	**Total** **abs. (%)**	**MH-RR [95% KI]**
PGA ≤ 2	4 (33.3%)	3 (23.1%)	7 (28.0%)	RR = 0.53 [0.14, 1.97], *p* = 0.342
PGA > 2	4 (33.3%)	7 (53.8%)	11 (44.0%)	
PGA missing at week 16	4 (33.3%)	3 (23.1%)	7 (28.0%)	
SAA ≤ ULN (10 mg/L)	2 (16.7%)	7 (53.8%)	9 (36.0%)	3.90 [1.12, 13.5], *p* = 0.015
SAA > ULN (10 mg/L)	6 (50.0%)	2 (15.4%)	8 (32.0%)	
SAA missing at week 16	4 (33.3%)	4 (30.8%)	8 (32.0%)	
CRP ≤ ULN (0.5 mg/dL)	3 (25.0%)	9 (69.2%)	12 (48·0%)	2.45 [1.09, 5.53], *p* = 0.011
CRP > ULN (0.5 mg/dL)	5 (41.7%)	1 (7.7%)	6 (24.0%)	
CRP missing at week 16	4 (33.3%)	3 (23.1%)	7 (28.0%)	
VAS physician, baseline, in mm	66.5 (0.0; 90.0)	60.0 (24.0; 82.0)	65.0 (0.0; 90.0)	
VAS physician, week 16, in mm	38.0 (0.0; 77.0)	29.5 (0.0; 62.0)	34.0 (0.0; 77.0)	0.18 (for the absolute difference at week 16)
VAS patient, baseline, in mm	47.0 (0.0; 77.0)	58.0 (15.0; 88.0)	55.5 (0.0; 88.0)	
VAS patient, week 16, in mm	38.5 (0.0; 78.0)	33.0 (0.0; 78.0)	33.5 (0.0; 78.0)	0.91 (for the absolute difference at week 16)
FFbH, baseline, in %	88.9 (8.3; 100)	91.7 (66.7; 100)	91.7 (8.3; 100)	
FFbH, week 16, in %	86.6 (58.8; 100)	91.7 (80.6; 100)	88.9 (58.8; 100)	0.11

**Table 3 jcm-11-05360-t003:** Line listing of adverse events showing no new safety concerns.

	Randomized Treatment Group		
AEs until Date of V6/Week 16	PBO (*n* = 12) (%)	TCZ (*n* = 13) (%)	Total (*n* = 25) (%)
Patients with AEs	8 (66.7%)	12 (92.3%)	20 (80%)
Number of AEs	22	51	73
No category	0	0	0
Infection	5 (22.7%)	10 (19.6%)	15 (20.5%)
Joint complaint	1 (4.5%)	6 (11.8%)	7 (9.6%)
FMF flare	4 (18.2%)	10 (19.6%)	14 (19.2%)
Skin disorder	2 (9.1%)	6 (11.8%)	8 (11.0%)
Cardiac	3 (13.6%)	2 (3.9%)	5 (6.8%)
Gastroenterology	2 (9.1%)	9 (17.6%)	11 (15.1%)
Other	5 (22.7%)	8 (15.7%)	13 (17.8%)

## Data Availability

The data presented in this study are available in the supplementary material or on request from the corresponding author.

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
