# Peer review of "Tocilizumab for the Treatment of Familial Mediterranean Fever—A Randomized, Double-Blind, Placebo-Controlled Phase II Study"

_jcm, 2022, doi:10.3390/jcm11185360_

Round 1
Reviewer 1 Report
Dear Authors,
I read with interest the manuscript "Tocilizumab for the treatment of Familial Mediterranean Fever - A randomized double blind placebo-controlled phase II study", which investigates the response and safety of crFMF to tocilizumab. Systematic standardized research data for response to IL-6 inhibition in FMF are urgently need. Therefore, study addresses a scientific knowledge gap. Furthermore, the manuscript is well written, for language editing I feel not competent. However, I have some comments and suggestions.
1) General
The authors use both terms "efficacy" (abstract, objective, introduction) and "effectiveness" (discussion) and at the same time for primary endpoint "complete remission" but use later the term “responder/response”. Since "effectiveness" and "efficacy" are not the same, I suggest to define which primary/secondary objective was chosen and align wording. Personally, I think the study was performed to investigate “effectiveness”. In addition, “response” and “remission” are from my point of view not the same. In several rheumatologic diseases remission is defined over a certain time period (clinical remission on medication = 6 months inactive disease, e.g. Wallace et al.). In this study primary endpoint is to analyze “the TCZ effect” at week 16, therefore I guess “response” represents the context better. This would be than also in line with the Cluster study which was used several times as reference.
Minor comment: up to my knowledge genes are written usually italic (MEFV gene).
2) Introduction
- Line 45/46 - why is FMF the prototype for monogenetic AID? I suggest to delete this phrase.
- For readers not familiar with FMF the relationship between elevated SAA and amyloidosis might not be clear. As SAA was used to define the primary endpoint, I suggest to explain this relationship in the introduction, to highlight the importance of inactive disease and why “complete response” and over the disease course “complete remission” is important.
- Line 58 I think it is colchicine resistance and not refractory, or? I suggest to use the up to date definition, see Ozen et al. 2021
- Line 65 The others are correct that currently no IL-1 studies are available showing prevention of amyloidosis, but several studies are available showing improvement of kidney function in patients with Amyloidosis after receiving treatment. Therefore, I recommend to rephrase the sentences.
- Line 68 Minor comment: juvenile idiopathic arthritis (but up to my knowledge only in SJIA and polyarthritis)
3) Materials and Methods
- Line 80 – Why did the authors did not use the “Gattorno” classification criteria, as those are validated and are recommended for clinical studies?
- Line 80 – I suggest to define which type of variants are included in the manuscript – pathogenic or likely pathogenic variants. I guess no VUS If I have screened the suppllementary correct, or?
- Line 81 – Please define “inadequate response” and “intolerance” – or in the supplementary material how “disease activity” was defined.
- Was subclinical inflammation (elevated CRP or SAA ) an inclusion criteria? I assume yes, because otherwise the primary endpoint definition “PGA + normalized ESR/CRP + SAA” might be not very robust, as usually no flare = no CRP or SAA. If subclinical inflammation was not an inclusion criteria, authors should comment on this.
- Line 86-94: Unfortunately, I was not able to understand the PGA definition – what did the others classify as “symptoms” did they mean febrile flares/attacks? I`m sorry but link from original trial did not work. In the supplementary Table S4 in the manuscript they have noted flare characteristics – how did authors handled that as FMF flares are usually 6-72 hours. Maybe it is possible to explain that as a Note in the supplementary.
- Line 152 – Was ethic approval really only needed from Tuebingen and not by local CECs for the participating study centers (= multicenter Trial)?
4) Results
- Table 1 – Minor comment: columne 1 has no homogeneous structure (sometimes “,” than “;” or “nothing”)
- Lines 172/173- Minor comment “end point” should be “endpoint”
- Lines 177-180 and table 4 - > In my opinion, the 7 patients who did not reach the primary endpoint are drop outs (prematurely study termination) -> thus, in my opinion, only 8 (placebo) or 10 (TCZ) patients reached the endpoint and should be used for the results (cave discrepancies table/ text - text N =7, table n= 8 “drop outs”). The 7 drop outs could/should then of course be analyzed as a subcohort at the time of the drop out, and that would be relevant – if all 3 patients with TCZ prematurely stopped study participation due to ineffectiveness/worsening of the disease course or safety events – this is a clinical important information. I suggest that authors maybe re-discuss and then also address that in the discussion section.
5) Discussion
Line 226/233 - is it not to be expected that the patients who are included as "non-responders to colchicine" do not show any response? That is why they were included...therefore I`m not sure if you really can speak from "superiority" – particularly as only 15% have shown “response”. I knew it is to late as study is performed and I assume authors have discussed why they have not used a “cross over” study design, but maybe they would like to address that in their limitation section.
Lines 263-264 - As the authors address, it is well known that TCZ can make "false negative CRP" - why didn't they determine S100 proteins for “Response”, particularly as this is a well-known marker in FMF to monitor disease activity? I think that should be discussed.
Lines 286/287 – Even if at study conduct the definitions from Ozen at al might not have been published the authors should have defined by themselves “colchicine resistance” as they used that as inclusion criteria (see comment methods section).
6) Conclusion
Lines 289/290 – From my point of view this message is very optimistic – 3 TCZ patients have not completed the trial due to ineffectiveness or AEs and only 2 patients met response criteria.
7) Supplementary
Did the authors have monitored the Liver enzymes – as this is a common site effect in TCZ?
Author Response
Dear Reviewer,
Thank you very much for your very helpful comments which we hope to answer satisfactorily. Please find my answers in yellow/bold and in the marked version of the manuscript.

Reviewer 2 Report
General comments
The authors performed a randomized, double-blinded, placebo-controlled phase II trial in adult patients with active FMF and inadequate response or intolerance to colchicine to evaluate the efficacy and safety of the IL-6 receptor antibody Tocilizumab (TCZ) over a 24 week-period of treatment with tocilizumab. The primary endpoint was the number of patients achieving complete remission at week 16, defined as a PGA ≤ 2 and normalized ESR or CRP inclusion normalized SAA.
In my opinion, the main clear results are not the superiority of tocilizumab over placebo on the higher number of patients achieving complete remission, as claimed by the authors , but rather a normalization of ESR and CRP in patients that have received tocilizumab.
Specific comments
My comments are as follows:
Abstract : line 29 : ESR and SAA should be detailed in the abstract
Line 36 : No few safety aspect occured : given the small number of patients enrolled, the safety aspect could not be evaluated. This point should be corrected.
Material and methodes
The authors should mentionned in materials and methods that written informed consents of patients were obtained by the investigators.
Line 98 : Could the authors detail the different periods of wash out that had to be respected and the concerned drugs ?
Line 146 : « Missing endpoints due to missing data were imputed as non responders » .
Could the authors explain why, and how this imutation could not have changed the results ?
Line 104 : Why do the authors fix the alpha risk to 0.2 with is much higher thant the usul alpha risk set to 0.05.
Result section
Glucocorticoids were used in 32%. However the authors indicated (line 96) that patients were excluded if they were ever treated with TCZ or glucocorticoide >10 mg/day. Please clarify the dose of GC of patients enrolled in the study.
Line 174 « Therefore, the superiority of TCZ 174 compared to placebo could be shown concerning the pre-specified significance level of α=0.2 (p=0.089, supplementary S8) ». The authors should attenuate their conclusion, as the number of patients enrolled and who present a positive outcome (two patients in TCZ group vs zero in placebo group) is very small and the alpha risk set to 0.2 instead of 0.05. The sentence of the discussion lines 226-227 should be rewritten to be in adequation with the results, and highlyting the very few number of patients and the unsual high alpha risk.
« Due to insufficient response or side effects, seven (28%) patients terminated earlier than week 16, which led to missing data at week 16. These early terminations occurred in four (33.3%) patients in the PBO arm and in three (23.1%) patients in the TCZ arm, respectively. These patients should not be excluded of the statistical analysis. » An intention to treat statistical analysis should be performed. If this was done, please clarify, since it is not clear in the paper.
Please indicate the number of outliers > 100mg/l that were excluded in this graph (figure 3) and in which group.
Why did the authors not apply statistical analyses to compare the frequency of adverse effects between placebo group and TCZ group ?
Please delete « treatment groupKomi » in figure 2 and figure 3.
Author Response
Dear Reviewer,
Thank you very much for your very helpful comments which we hope to answer satisfactorily. Please find my answers in bold and yellow and in the marked version of the manuscript.
